

# Topological order in matrix Ising models

**Sean A. Hartnoll, Edward A. Mazenc and Zhengyan D. Shi**

Department of Physics, Stanford University, Stanford, CA 94305-4060, USA

## Abstract

We study a family of models for an $N_1 \times N_2$ matrix worth of Ising spins $S_{aB}$. In the large $N_i$ limit we show that the spins soften, so that the partition function is described by a bosonic matrix integral with a single 'spherical' constraint. In this way we generalize the results of [1] to a wide class of Ising Hamiltonians with $O(N_1, \mathbb{Z}) \times O(N_2, \mathbb{Z})$ symmetry. The models can undergo topological large $N$ phase transitions in which the thermal expectation value of the distribution of singular values of the matrix $S_{aB}$ becomes disconnected. This topological transition competes with low temperature glassy and magnetically ordered phases.

# 1 Overview

Some years ago now, a remarkable work introduced a model of non-locally interacting Ising spins whose high temperature phase could be mapped onto a matrix integral, allowing the partition function to be computed [1]. The original interest in this model was due to the fact that the low temperature phase — not captured by a matrix integral — described a structural glass. Our objective in this paper is twofold. Firstly, we will generalize the solution of the model of [1] to several families of $N_1 \times N_2$ non-locally interacting spins. Secondly, we will emphasize that, prior to vitrification, these models can generically undergo topological large $N$ phase transitions. Such transitions are known to be ubiquitous in matrix integrals, the Gross-Witten-Wadia transition being a well-known example [2,3], but are nontrivial from the perspective of the original Ising spins. The connectivity of the large $N$ singular value distribution of a matrix of Ising spins gives a simple instance of topological order in a classical spin system.

The heart of the first result is a spin softening theorem, showing that the discreteness of the Ising spin variables is (almost) washed away in the large $N_i$ limit. The variables no longer square to unity and a single 'spherical constraint' on the emergent bosonic degrees of freedom remains. This is a well-established phenomenon in spin models [4,5]. More precisely, given an $N_1 \times N_2$ matrix worth of Ising spins $S_{aB} \in \pm 1$, we will show that for certain classes of spin Hamiltonians $H[S]$, at temperatures above any glassy or ordering transitions, the partition function

$$\sum_{S_{aB}=\pm 1} e^{-\beta H[S]} \xrightarrow{N_i \to \infty} \left(2e^{-\frac{1}{2}}\right)^{N_1 N_2} \int dM \delta(\text{tr}[MM^T] - N_1 N_2) e^{-\beta H[M]}. \tag{1.1}$$

Here $M_{aB}$ is a matrix of bosons. The configuration space of the spins are the $2^{N_1 N_2}$ vertices of a hypercube, while the bosons take values in a hypersphere $S^{N_1 N_2 - 1}$. The bosonic integrals can be evaluated using standard techniques [6].

We will focus on the family of Hamiltonians

$$H = \sum_n \frac{v_n}{N_1^{n-1}} \text{tr}\left[(SS^T)^n\right] \equiv \text{tr}\left[V(SS^T)\right], \tag{1.2}$$

where the trace tr is over the matrix indices and the $v_n$'s are order one couplings. The model with the $n = 2$ term only, which is quartic in the spins, was mapped to matrices in [1,7] using a Hubbard-Stratonovich decoupling — familiar from replica descriptions of disordered spins [8] — as a key step. In §2 we generalize those arguments to terms with $n > 2$. The essential characteristic of the Hamiltonian (1.2) is not the matrix-like interactions, but rather the $O(N_1, \mathbb{Z}) \times O(N_2, \mathbb{Z})$ symmetry (described in [9]). For example, our spin softening theorem also applies to models of the form $H = \sum_n \frac{u_n}{N^n} \sum_{a \neq b} [(SS^T)_{ab}]^{2n}$.

To make the spin softening (1.1) tangible, Fig.1 contains the results of numerical simulations of the spin system (1.2) with $N_1 = N_2 = 120$ together with the large $N_i$ matrix integral result. Two illustrative cases are plotted, $H = \text{tr}\left[(SS^T)^3\right]$ and $H = -3\text{tr}\left[(SS^T)^4\right] + \text{tr}\left[(SS^T)^5\right]$. The former is the next simplest monomial potential, beyond the $n = 2$ case studied in [1]. The latter, as we shall see, illustrates how negative terms in the Hamiltonian can induce topological transitions. The energy $E = -\partial(\log Z)/\partial\beta$ is seen to match up to $1/N_i$ corrections, as advertised, above a glassy transition temperature $T_{gl}$. Below the glassy temperature, the matrix model energy continues to decrease while the Ising model 'freezes out' [1]. In the plots, we have also marked with a dot the location of the topological transition. These transitions occur prior to the glassy freeze-out and are hence captured by the matrix integral. In §3 we give a detailed description of these third order transitions by solving the matrix integral. In §4 we show how the change in connectivity of the distribution of singular values of the $S_{aB}$ spin matrices can be seen clearly in numerics, even while the non-analyticity in the energy as

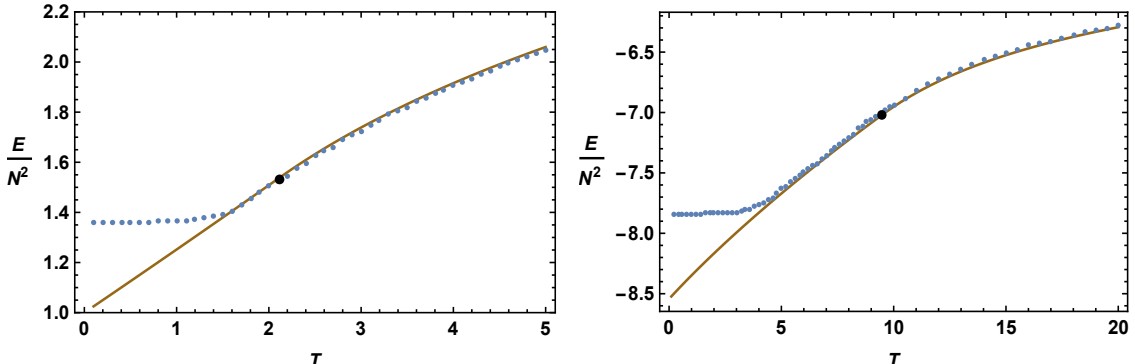

Figure 1: Large $N$ energy density of two matrix Ising models as a function of temperature computed by numerical Monte Carlo simulation of spins (blue dots) and analytically from the corresponding matrix model (brown curve). The left plot has $H = \mathrm{tr}\big[(SS^T)^3\big]$ and the right plot has $H = -3\,\mathrm{tr}\big[(SS^T)^4\big] + \mathrm{tr}\big[(SS^T)^5\big]$. The black dot indicates the location of the topological transition, which is above the glassy transition in both cases.

a function of temperature is very weak. We also describe a finite $N$ approximation to the large $N$ topological order parameter (the number of components of the distribution) that makes the critical temperature identifiable in numerical simulations of the spin system.

In the discussion in §5 we comment on the importance of topological phase transitions for generalizing the spin softening results to quantum matrix spin systems.

## 2 Proof of spin softening

In this section, we give a rigorous derivation of 'spin softening', focusing on models of the form (1.2). Many steps are similar to those in [1], with differences due to the fact that a general potential $V(SS^T)$ cannot be mapped to a Gaussian integral via a Hubbard-Stratonovich transformation.

The strategy can be outlined as follows. First, we trivially rewrite the sum over spin values as $N_1 N_2$ constrained integrals. Inserting multiple resolutions of the identity, we introduce the collective field $G_{ab} = (SS^T)_{ab}$ as well as a Lagrange multiplier field $\sigma_{ab}$. We then show how only $O(N_1, \mathbb{R})$ singlets contribute at large $N_i$. The resulting path integral is then seen to be identical to the corresponding $G, \sigma$ integrals for a matrix model with continuous entries $M_{aB}$ and a single spherical constraint. After integrating the collective fields back out, we arrive at the promised spherically constrained matrix model.

The $G, \sigma$ fields are introduced as follows:

$$
\begin{aligned}
Z(\beta) = \mathrm{Tr}\, e^{-\beta H} &= \int dS \delta(S_{aB}^2 - 1) e^{-\beta\,\mathrm{tr}\big[V(SS^T)\big]} \\
&= \int dG \int dS \delta(G_{cd} - (SS^T)_{cd}) \delta(S_{aB}^2 - 1) e^{-\beta\,\mathrm{tr}[V(G)]} \\
&= \int dG\, e^{-\beta\,\mathrm{tr}[V(G)]} \int \frac{d\sigma}{2\pi} \int dS \delta(S_{aB}^2 - 1) e^{-i\sum_{cd}\sigma_{cd}(G_{cd} - (SS^T)_{cd})} \\
&= \int dG\, e^{-\beta\,\mathrm{tr}[V(G)]} \int \frac{d\sigma}{2\pi} e^{-i\sum_{cd}\sigma_{cd}G_{cd}}\, \mathrm{Tr}\, e^{i\sum_{ab}\sigma_{ab}(SS^T)_{ab}}.
\end{aligned}
\tag{2.1}
$$

In the last line, we have rewritten the $S$ integral again as a trace over spin operators (not over

matrix indices). The next step will be to compute this trace.

The first step in evaluating the trace, following [1], is to introduce an undetermined set of variables $\mu_a$ by adding zero to the exponent in the trace as $0 = \sum_a \mu_a(N_2 - (SS^T)_{aa})$. With this additional term, we can write (this step is where $O(N_2, \mathbb{Z})$ symmetry is being used)

$$\operatorname{Tr} e^{i \sum_{ab} \sigma_{ab}(SS^T)_{ab}} = e^{iN_2 \sum_a \mu_a} z(\sigma, \mu)^{N_2}, \tag{2.2}$$

where, following some standard manipulations [8]

$$z(\sigma, \mu) = \frac{1}{\sqrt{\det \tilde{\sigma}}} \int dw e^{-\frac{1}{2} \sum_{ab} w_a (\tilde{\sigma}^{-1})_{ab} w_b + \sum_a \log(2\cosh(w_a))} \tag{2.3}$$

$$= \frac{2^{N_1}}{\sqrt{\det \tilde{\sigma}}} \int dw e^{-\frac{1}{2} \sum_{ab} w_a [(\tilde{\sigma}^{-1})_{ab} - \delta_{ab}] w_b + \sum_a \left(-\frac{1}{12} w_a^4 + \frac{1}{45} w_a^6 + \cdots\right)}. \tag{2.4}$$

Here, we have defined a new variable $\tilde{\sigma}_{ab} \equiv 2i(\sigma_{ab} - \mu_a \delta_{ab})$. Using (2.2) and (2.3) in (2.1), we see that there are no sums over spins left, only bosonic integrals. However, while the powers of $w_a$ in (3.21) that are greater than two are invariant under $O(N_1, \mathbb{Z})$ transformations $w_a \to O_{ab} w_b$, they are not invariant under continuous $O(N_1, \mathbb{R})$ transformations. The crucial step in the spin-softening theorem is now to show that a certain choice of the $\mu_a$ (thus far arbitrary) renders these non-singlet terms negligible in the large $N_i$ limit.

The propagator for the $w_a$ in (2.4) is seen to be $P_{ab}(\tilde{\sigma}) \equiv (1/(\tilde{\sigma}^{-1} - 1))_{ab} = (\tilde{\sigma}/(1 - \tilde{\sigma}))_{ab}$. A sufficient condition for the non-singlet terms to be negligible at large $N$ is that

$$P_{ab}(\tilde{\sigma}) = O\left(1/\sqrt{N}\right) \quad \forall a \neq b \qquad \text{and} \qquad P(\tilde{\sigma})_{aa} = 0 \quad \forall a. \tag{2.5}$$

This can be verified by expanding the exponential, Wick contracting, and re-exponentiating (see [7] for a more explicit discussion). We can now check that the first set of conditions in (2.5) are automatically true while the latter are not. This second set of $N$ conditions can be imposed, however, by a suitable choice of the $N$ quantities $\mu_a$. This amounts to setting $\mu_a = \mu_a^\star$ such that

$$\left(\frac{1}{1 - \tilde{\sigma}^\star}\right)_{aa} = 1 \qquad \forall a. \tag{2.6}$$

Here $\tilde{\sigma}_{ab}^\star \equiv 2i(\sigma_{ab} - \mu_a^\star \delta_{ab})$. It remains, then, to verify the first set of conditions in (2.5).

*Assuming* that the scaling of the components of $G$ with $N$ is determined by the matrix integral term in the last line of (2.1), we can establish that the variance $\Delta G_{ab} \sim \sqrt{N}$ by standard random matrix theory arguments. It then follows from (2.1) that $\Delta \sigma_{ab} \sim 1/\Delta G_{ab} \sim 1/\sqrt{N}$, and therefore $\Delta P_{ab} \sim 1/\sqrt{N}$ for $a \neq b$, while $\Delta P_{aa} \sim 1$. A more rigorous derivation of these statements is given in Appendix A. These variances give the typical contribution of components of the propagator $P$ to the integral (2.4). The $a \neq b$ components are of the magnitude required by (2.5), while the diagonal terms are too large. For this reason, the constraint (2.6) must be imposed. Imposing this condition, we proceed to drop the non-singlet terms in (2.4). While the assumption of matrix scaling of $G$ is self-consistent, we will see in §4.3 that it fails to capture glassy or magnetically ordered regimes at low temperatures.

After dropping the non-singlet terms in (2.4), simple manipulations (doing the $w$ integral, simplifying the determinants, and introducing a new integral over a matrix $M$) lead to

$$\operatorname{Tr} e^{i \sum_{ab} \sigma_{ab}(SS^T)_{ab}} = 2^{N_1 N_2} e^{iN_2 \sum_a \mu_a^\star} \int dM e^{-\frac{1}{2} \sum_{abC} M_{aC}(1 - \tilde{\sigma}^\star)_{ab} M_{bC}} \tag{2.7}$$

$$= 2^{N_1 N_2} \int d\mu e^{iN_2 \sum_a \mu_a} \int dM e^{-\frac{1}{2} \sum_{abC} M_{aC}(1 - \tilde{\sigma})_{ab} M_{bC}}. \tag{2.8}$$

In the second line, we used the remarkable — and greatly simplifying — fact that the value $\mu^\star$ required for (2.6) is precisely the value picked out as the large $N$ saddle point if $\mu$ is integrated over. This allows us to avoid needing to find $\mu^\star$ explicitly as a function of $\sigma$.

Using (2.8) in (2.1), we can do the $\sigma$ integral (obtaining a delta function) and then the $G$ integral (which 'eats up' the delta function) to obtain

$$Z(\beta) = 2^{N_1 N_2} \int dM \int d\mu e^{i \sum_a \mu_a [N_2 - (MM^T)_{aa}]} e^{-\frac{1}{2} \mathrm{tr}[MM^T]} e^{-\beta \mathrm{tr}[V(MM^T)]}. \qquad (2.9)$$

In (2.9), the microscopic $N_1 N_2$ constraints $(S_{aB})^2 = 1$ have been reduced to the $N_1$ constraints $\sum_A (M_{aA})^2 = N_2$, imposed by the Lagrange multipliers $\mu_a$. To make further progress, we argue that a consistent large $N$ saddle point has $\mu_a = \mu$ for all $a$. This is true because upon integrating out $M$ to get an effective action for the $\mu_a$, the large $N$ saddle point equations for $\mu_a$ are permutation invariant. *Assuming* that this is the dominant large $N$ saddle, we finally obtain:

$$Z(\beta) = \left(2 e^{-\frac{1}{2}}\right)^{N_1 N_2} \int dM \int d\mu e^{i\mu[N_1 N_2 - \mathrm{tr}(MM^T)]} e^{-\beta \mathrm{tr}[V(MM^T)]}. \qquad (2.10)$$

This is the 'spin softened' partition function advertized in (1.1) and seen in the numerical results of Fig.1. We have also verified numerically that the partition functions (2.9) and (2.10) agree at all temperatures, justifying this last assumption a posteriori.

When the matrix integral correctly captures the large $N$ spin partition function, it will also capture connected correlators of spins of the form $\left\langle \mathrm{tr}\left[(SS^T)^{k_1}\right] \cdots \mathrm{tr}\left[(SS^T)^{k_n}\right] \right\rangle_c$. These are obtained by introducing sources $J_k \mathrm{tr}\left[(SS^T)^k\right]$ into the action and differentiating the partition function with respect to the couplings $J_k$. Thus, for example, the energy $E = -\partial_\beta \log Z$ and specific heat $C = -\partial_\beta^2 \log Z$ are captured by the matrix integral. On the other hand, non-singlet observables such as the magnetization $M = \left\langle \sum_{aB} S_{aB} \right\rangle$ and the susceptibility $\chi = \left\langle \sum_{aB} S_{aB} \sum_{cD} S_{cD} \right\rangle_c$ are not captured by the matrix description (as can be verified numerically).

In Appendix B we show that this spin softening theorem also goes through for the class of Hamiltonians

$$H = \sum_n \frac{u_n}{N^n} \sum_{a \neq b} [(SS^T)_{ab}]^{2n} = U(SS^T). \qquad (2.11)$$

## 3 Topological transition in the large $N$ matrix integral

### 3.1 The distribution of singular values

The partition function (2.10) can be computed using standard methods for matrix integrals. The matrix $M$ admits a singular value decomposition

$$M = U \Lambda V^T, \qquad (3.1)$$

where $U$ and $V$ are orthogonal matrices and $\Lambda$ is the diagonal matrix formed out of the singular values $\{\lambda_i\}$ of $M$. The matrix integral in (2.10) does not depend on the angular variables $U$ and $V$, so these integrals can be performed trivially. We will further restrict attention to the case of square matrices with $N_1 = N_2$.[1] The measure $dM = J dU dV d\Lambda$, with the Jacobian

---

[1] When $N_1 \neq N_2$ there is an extra $\log|\lambda_i|$ term in the effective action for the singular values [1,10]. This term gives a repulsive force away from the origin and causes the distribution of singular values to be disconnected, even at high temperatures. Topological transitions can still occur in such cases, along the lines of the **1 → 3** transition considered below.

$J = \prod_{i<j} |\lambda_i^2 - \lambda_j^2|$ as in [1]. Finally, we introduce the rescaled variables $\sqrt{N} x_i = \lambda_i$ to write

$$Z(\beta) = \text{const} \cdot \int d\mu \int \left(\prod_i dx_i\right) e^{N^2 \left[ i\mu\left(1 - \frac{1}{N}\sum_i x_i^2\right) - \beta \frac{1}{N}\sum_i \hat{V}(x_i) + \frac{1}{2}\frac{1}{N^2}\sum_{i\neq j} \log|x_i^2 - x_j^2|\right]}. \tag{3.2}$$

Here, from (1.2),

$$\hat{V}(x) = \sum_n v_n x^{2n}. \tag{3.3}$$

On the saddle point $i\mu$ will be real, and so we set $i\mu \equiv \hat{\mu}$ in the following.

At large $N$, the integrals in (3.2) can be evaluated on the saddle point. The two saddle point equations are

$$\frac{1}{N}\sum_i x_i^2 = 1, \qquad \hat{\mu} x_i + \frac{\beta}{2}\hat{V}'(x_i) - \frac{1}{N}\sum_{j\neq i} \frac{x_i}{x_i^2 - x_j^2} = 0. \tag{3.4}$$

In terms of the normalized and symmetrized density of singular values,

$$\rho(x) = \frac{1}{2N}\sum_i \left[\delta(x - x_i) + \delta(x + x_i)\right], \tag{3.5}$$

the saddle point equations can be written as the integral equations:

$$\int dx \rho(x) x^2 = 1, \qquad \hat{\mu} x + \frac{\beta}{2}\hat{V}'(x) = P\int dy \rho(y) \frac{1}{x - y}. \tag{3.6}$$

The second equation in (3.6) describes the singular values moving in an external potential

$$V_{\text{ext}}(x) = \frac{1}{2}\left(\beta \hat{V}(x) + \hat{\mu} x^2\right), \tag{3.7}$$

and with a logarithmic repulsive interaction between them. In the high temperature limit ($\beta \to 0$) the quadratic $\hat{\mu} x^2$ term dominates the external potential $V_{\text{ext}}(x)$. One finds that $\hat{\mu} \to \frac{1}{2}$. The balance between the quadratic external potential and the logarithmic repulsion leads to the well-known connected Wigner semi-circle distribution. In the low temperature limit ($\beta \to \infty$), the external potential becomes strong and overcomes the logarithmic repulsion. The singular values accumulate at the minima $x_\star$ of the external potential: $\rho(x) \to \sum_\star s_\star \delta(x - x_\star)$. We will proceed to show that in all cases the external potential $V_{\text{ext}}(x)$ develops minima away from the origin, and therefore the low temperature distribution is disconnected. This necessitates a topological transition at intermediate temperatures.

## 3.2 Potentials with a unique minimum and the $1 \to 2$ transition

In this subsection, we consider the case of potentials $\hat{V}(x)$ with a unique minimum at the origin. A disconnected distribution arises at low temperatures because the constraint $\int dx \rho(x) x^2 = 1$ does not allow all the singular values to collapse to zero. This translates into $\hat{\mu} < 0$ in the external potential (3.7) at low temperatures. The external potential now has a pair of minima at $x = \pm x_\star$, leading to a distribution with two disconnected components at low temperatures: $\rho(x) \to \frac{1}{2}(\delta(x - x_\star) + \delta(x + x_\star))$. The constraint $\int dx x^2 \rho(x) = 1$ then fixes $x_\star = 1$, and hence $\hat{\mu} \to -\frac{1}{2}\beta \hat{V}'(1)$. The energy (3.18) of this zero temperature state is $E = N^2 \hat{V}(1)$. For this class of potentials, therefore, we expect a transition from $1 \to 2$ components at intermediate temperatures. We proceed to characterize this transition in detail. In the following subsection, we will consider the case where $\hat{V}(x)$ already has additional minima, prior to consideration of the constraint.

The second integral equation in (3.6) can be solved using well-known methods [6]. In particular, the connected 'single-cut' solution can be written in the form

$$\rho(x) = \frac{\hat{\mu}}{\pi}\sqrt{a^2 - x^2} - \sum_n \frac{\beta v_n}{\pi}\frac{(2n)!}{4^n[(n-1)!]^2}\frac{x^{2n}}{|x|}B\left(\frac{x^2}{a^2}, \frac{1}{2}-n, \frac{1}{2}\right),$$ (3.8)

with support on $[-a, a]$. Here, $B$ denotes an incomplete beta function. The distribution has the form of a polynomial times $\sqrt{a^2 - x^2}$. Given the solution (3.8), the two constants $a$ and $\hat{\mu}$ are determined by imposing $\int dx\rho(x) = 1$ and $\int dx\rho(x)x^2 = 1$. The integrals can be done explicitly, and the constraints become

$$\frac{\hat{\mu}a^2}{2} + \sum_n \beta v_n \frac{a^{2n}(2n)!}{4^n n!(n-1)!} = 1, \qquad \frac{\hat{\mu}a^4}{8} + \sum_n \beta v_n \frac{na^{2n+2}(2n)!}{2\cdot 4^n(n+1)!(n-1)!} = 1.$$ (3.9)

In solving the constraint equations, it is important to restrict to solutions where the distribution $\rho(x)$ is everywhere non-negative.

At some critical $\beta_c$, a solution to the constraints (3.9) leads to a zero in the distribution. For $\beta > \beta_c$ (i.e. at low temperatures), the single-cut solution will no longer be non-negative everywhere and the correct solution is necessarily disconnected. From the physical discussion of the external potential above, it is clear that the distribution will disconnect at the origin. Therefore, the critical temperature can be determined from the condition that $\rho(0) = 0$:

$$\hat{\mu}a^2 + \sum_n \beta_c v_n \frac{a^{2n}(2n)!}{4^n(n-\frac{1}{2})[(n-1)!]^2} = 0.$$ (3.10)

For example, in the case of a monomial potential $\hat{V}(x) = v_n x^{2n}$ we can solve (3.9) and (3.10) explicitly to obtain the critical temperature

$$T_c = \frac{1}{\beta_c} = \frac{v_n}{2\sqrt{\pi}}\left[\frac{4}{3}\left(1+\frac{1}{n}\right)\right]^n \frac{\Gamma\left(n-\frac{1}{2}\right)}{\Gamma(n-1)}.$$ (3.11)

Within this class of models, $T_c/v_n$ increases monotonically from $T_c = v_2$ at $n = 2$ to

$$T_c \sim v_n\sqrt{\frac{ne^2}{4\pi}}\left(\frac{4}{3}\right)^n \qquad \text{as} \qquad n \to \infty.$$ (3.12)

In this limit the critical temperature increases exponentially with $n$. The width of the distribution at the critical point in these models is $a^2 = \frac{4}{3}\frac{1+n}{n}$, which remains finite as $n \to \infty$. It is simple to determine the critical temperature numerically for more general models with polynomial potentials (but still with a single minimum, at the origin).

Once a connected distribution of singular values ceases to exist, one must look for a disconnected 'two cut' solution. The solution can be found as in e.g. [11], and can be written as

$$\rho(x) = \frac{1}{\pi}\sum_n \beta v_n Q_n(x)\sqrt{(b^2 - x^2)(x^2 - a^2)},$$ (3.13)

with support on $[-b, -a] \cup [a, b]$ where the polynomial

$$Q_n(x) = n|x|^{2n-3}\sum_{p=0}^{n-2}\frac{b^{2p}}{(2x)^{2p}}\frac{(2p)!}{(p!)^2}\,{}_2F_1\left(\frac{1}{2}, -p, \frac{1}{2}-p; \frac{a^2}{b^2}\right).$$ (3.14)

The constants $a$, $b$ and $\hat{\mu}$ are determined through the two constraints

$$1 = \frac{\hat{\mu}}{2}(a^2 + b^2) + \sum_n \beta v_n \frac{b^{2n}(2n)!}{4^n n!(n-1)!}\,{}_2F_1\left(\frac{1}{2}, -n, \frac{1}{2}-n; \frac{a^2}{b^2}\right),$$ (3.15)

$$0 = \hat{\mu}b^2 + \sum_n \beta v_n \frac{n}{n-\frac{1}{2}}\frac{b^{2n}(2n)!}{4^n n!(n-1)!}\,{}_2F_1\left(\frac{1}{2}, 1-n, \frac{3}{2}-n; \frac{a^2}{b^2}\right),$$ (3.16)

as well as the condition that $\int dx \rho(x) x^2 = 1$. This last integral can be done in closed form and the constraint becomes

$$1 = \sum_n \frac{2n b^{2n+2}}{2^{2n+2}} \beta v_n \sum_{p=0}^{n-2} \frac{(2p)!(2(n-p))!}{(n-p)!(1+n-p)!(p!)^2} \times$$
$$_2F_1\left(-\frac{1}{2}, -1-n+p, \frac{1}{2}-n+p, \frac{a^2}{b^2}\right) {}_2F_1\left(\frac{1}{2}, -p, \frac{1}{2}-p, \frac{a^2}{b^2}\right). \qquad (3.17)$$

The appearance of a disconnected singular value distribution at $\beta_c$ leads to a third order large $N$ quantum phase transition [2,3,11]. We can see this explicitly as follows. The energy is given by

$$E = -\frac{d \log Z}{d\beta} = N^2 \int dx \rho(x) \hat{V}(x). \qquad (3.18)$$

This integral is easily evaluated on the single cut solution. It can also be evaluated on the two cut solution, in terms of sums of hypergeometric functions, similarly to (3.17). For the case of a monomial potential $\hat{V}(x) = v_n x^{2n}$, with critical temperature $T_c$ given by (3.11), the energy just above and just below the transition is thereby found to be

$$\frac{E}{T_c} = \begin{cases} \frac{1-4n^2}{2n(1-n^2)} + \frac{(1-2n)^2}{2n(1+n)^2} \frac{T-T_c}{T_c} - \frac{3(1-2n)^2}{4n(1+n)^3} \frac{(T-T_c)^2}{T_c^2} + \cdots & T > T_c \\ \frac{1-4n^2}{2n(1-n^2)} + \frac{(1-2n)^2}{2n(1+n)^2} \frac{T-T_c}{T_c} - \frac{(1-2n)^2}{2n(1+n)^2} \frac{(T-T_c)^2}{T_c^2} + \cdots & T < T_c \end{cases}. \qquad (3.19)$$

The second derivative of the energy with respect to temperature is seen to be discontinuous at the critical temperature. There is no symmetry breaking associated to this phase transition. It is a topological transition with a topological order parameter given by the number of components of the large $N$ distribution. We will discuss this order parameter further in §4.2 below.

## 3.3 Potentials with several minima and the $1 \to 3$ transition

When the potential had a single minimum, the topological transition was driven purely by the spherical constraint. This constraint prevented the singular values from accumulating at the origin at low temperatures. When the potential has several minima, however, there are minima away from zero already in $\hat{V}(x)$. This leads to a slightly different topological transition. For concreteness, we will focus on models with two terms such that

$$\hat{V}(x) = -|v_n| x^{2n} + v_{n+1} x^{2n+2}. \qquad (3.20)$$

Here, we choose $v_{n+1} > 0$ so that the function indeed has a pair of minima away from the origin. To see what kind of topological transition is expected to arise, we can solve for the low temperature distribution. The external potential will overcome the repulsion between singular values (as previously in the $1 \to 2$ transition), and so we look for a distribution of the form

$$\rho(x) = (1-2s_\star)\delta(x) + s_\star \left[\delta(x-x_\star) + \delta(x+x_\star)\right]. \qquad (3.21)$$

We are now allowing for some singular values to be at the origin because we will see shortly that $\hat{\mu} = -\frac{\beta}{2x_\star} V'(x_\star) > 0$ at low temperatures. The constraint $\int dx\, x^2 \rho(x) = 1$ implies that $x_\star^2 = 1/(2s_\star)$. The fraction $s_\star$ of singular values away from the origin is determined by minimizing the total energy $E = N^2 \int dx \rho(x) \hat{V}(x)$. This gives

$$x_\star^2 = \frac{(n-1)|v_n|}{n v_{n+1}}. \qquad (3.22)$$

This solution is valid so long as $x_\star \geq 1$, ensuring that the weight of the delta function at the origin is positive. When this condition is not satisfied, the energy is minimized by setting $x_\star = 1$, and there is no delta function at the origin. This case reduces to that in the previous section. However, when $x_\star > 1$, the zero temperature distribution has three connected components as in (3.21). This leads us to anticipate – in these cases – a topological transition $\mathbf{1} \to \mathbf{3}$ in which the high temperature connected distribution breaks into three separate components. We proceed to consider this case in more detail.

The high temperature distribution is again given by (3.8). However, the transition now occurs at $\beta_c = 1/T_c$ such that there is a point $x_c$ (typically away from the origin) with $\rho(x_c) = \rho'(x_c) = 0$. These two equations can be solved (e.g. numerically) for $\beta_c$ and $x_c$.

Below the critical temperature, the distribution takes the three-cut form

$$\rho(x) = \frac{1}{\pi} Q(|x|) \operatorname{sgn}(x^2 - a^2) \sqrt{(x^2 - a^2)(x^2 - b^2)(c^2 - x^2)}, \tag{3.23}$$

supported on $[-c, -b] \cup [-a, a] \cup [b, c]$, and with $Q(x)$ a degree $2n - 1$ polynomial that is odd under $x \to -x$. The presence of three cuts implies that the fraction of singular values in each cut is no longer fixed by symmetry. Following the discussion of [12], we introduce the extended action

$$S[\rho; f_\alpha, \Gamma_\alpha, \mu] = \int dx \Big( -\beta \hat{V}(x) + \frac{1}{2} \int dx' \rho(x') \log |x^2 - x'^2| \Big) \rho(x)$$
$$+ \sum_{\alpha=1}^3 \Gamma_\alpha [f_\alpha - \int_{C_\alpha} dx \rho(x)] + i\mu [1 - \int \rho(x) x^2 dx]. \tag{3.24}$$

This is the action (3.2), together with Lagrange multipliers $\Gamma_\alpha$ enforcing the filling fraction constraints $f_\alpha = \int_{C_\alpha} dx \rho(x)$. Here, $C_\alpha$ denotes the three disconnected supports in order of increasing $x$.

Due to the normalization constraint $\sum_\alpha f_\alpha = 1$ and the symmetry constraint $f_1 = f_3$, we can rewrite the action functional in terms of $f_2$ alone:

$$S[\rho; f_2, \Gamma_\alpha, \mu] = \int dx \Big( -\beta \hat{V}(x) + \frac{1}{2} \int dx' \rho(x') \log |x^2 - x'^2| \Big) \rho(x) \tag{3.25}$$
$$+ (\Gamma_1 + \Gamma_3)[1 - f_2 - \int_{C_1 \cup C_3} dx \rho(x)] + \Gamma_2 [f_2 - \int_{C_2} dx \rho(x)] + i\mu [1 - \int \rho(x) x^2 dx].$$

In order to solve for $\rho$, we now minimize (3.25) with respect to all of its arguments. The condition $\frac{\partial S}{\partial f_2} = 0$ gives

$$\Gamma_3 - \Gamma_2 = 0 = \int_a^b Q(x) \sqrt{(x^2 - a^2)(x^2 - b^2)(x^2 - c^2)}. \tag{3.26}$$

The $n$ polynomial coefficients of $Q(x)$ and the parameters $a, b, c, i\mu = \hat{\mu}$ are fixed by (3.26) together with the $n + 3$ constraints that follow from imposing the asymptotic behavior of the resolvent

$$\frac{1}{2} V'_{\text{ext}}(x) - Q(x) \sqrt{(x^2 - a^2)(x^2 - b^2)(x^2 - c^2)} \sim \frac{1}{x} + \frac{1}{x^3} \quad \text{as} \quad x \to +\infty. \tag{3.27}$$

The leading $1/x$ behavior is familiar from standard cases (see e.g. [6]). The subleading $1/x^3$ behavior is equivalent to imposing the spherical constraint that $\int \rho(x) x^2 dx = 1$. This follows from expanding the resolvent

$$G(x) \equiv \frac{1}{Nx} \left\langle \operatorname{tr} \left( 1 - \frac{M^T M}{Nx^2} \right)^{-1} \right\rangle = \frac{1}{x} + \frac{\langle \operatorname{tr}(M^T M) \rangle}{N^2 x^2} + \cdots = \frac{1}{x} + \frac{1}{x^3} + \cdots. \tag{3.28}$$

These constraints can be solved numerically and the large $N$ three cut distribution can be determined. In Fig. 2 we have seen that the matrix integral result obtained in this way matches the Monte Carlo simulation of Ising spins, below the topological transition temperature and above the glass transition temperature. The topological transition leads to a third order non-analyticity in the energy at $T_c$, similarly to the $\mathbf{1 \to 2}$ case discussed in the previous subsection.

# 4 Topological transition in the matrix Ising model

## 4.1 Numerical results

The matrix spin system can be simulated numerically using standard annealed Monte Carlo methods. The output is a thermal ensemble of matrices $S_{\alpha\beta}$ of spins. These matrices can be used to compute the thermal expectation value of the energy (1.2). Furthermore, the singular values of these matrices can be computed and binned to obtain a thermally averaged symmetrized distribution of singular values.

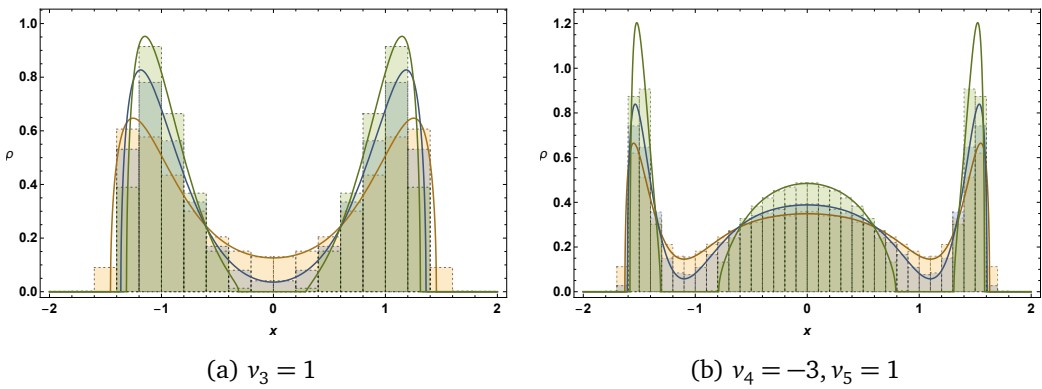

(a) $\nu_3 = 1$        (b) $\nu_4 = -3, \nu_5 = 1$

Figure 2: Distribution of singular values for two large $N$ matrix Ising models. Histograms are from Monte Carlo simulations of the matrix Ising model. Solid lines are analytically computed distributions from the corresponding matrix integral. For the model considered in the left plot (a), $T_c = 2.11$ and the distribution is shown at $T = 5, 2.6, 1.8$. For the right plot (b), $T_c = 9.52$ and the temperature shown are $T = 20, 12, 6$.

Fig. 2 shows the numerically computed symmetrized singular value distribution for the models whose energy was shown in Fig. 1 above. The figure shows excellent agreement with the matrix integral distribution, obtained by solving the matrix integral as described in the previous section (both single and multi-cut solutions). The figure also clearly reveals the topological transition, which is not obvious in Fig. 1.

## 4.2 Topological Order Parameter

The $N = \infty$ topological transition is characterized by a change in connectedness of the (symmetrized) distribution of singular values. Let $\rho(z)$ be an analytic continuation of this distribution to the complex plane. Then, an integer quantity that jumps across the topological transition is

$$n = \frac{1}{\pi i} \oint_\Gamma \frac{\rho'(z)}{\rho(z)} dz \,. \tag{4.1}$$

Here, $\Gamma$ is a contour that runs above and below the real axis. We saw in §3 that $\rho(z)$ has a square root branch cut on the real axis along the support of the solution and no further

zeros on the real axis. It follows that $n$ counts the number of disconnected components of the distribution on the real axis.

At large but finite $N$, the notion of connectedness of the distribution is not precisely defined, as the distribution is simply a sum of delta functions. Correspondingly, the transition will be smooth. In practice, however, at large but finite $N$ one can clearly see the topological transition in numerical simulations, as in Fig. 2 above. To define an 'order parameter' that approximates (4.1) and captures these changes at finite $N$, one must introduce a smeared version of the distribution

$$\rho_N(z) = \frac{\epsilon_N}{\pi N} \sum_{a=1}^{N} \frac{1}{(z-\lambda_a)^2 + \epsilon_N^2} = \frac{i}{2\pi N} \sum_{a=1}^{N} \left( \frac{1}{z - \lambda_a + i\epsilon_N} - \frac{1}{z - \lambda_a - i\epsilon_N} \right). \qquad (4.2)$$

The small number $\epsilon_N$ will be specified shortly. This function is not quite ready to be inserted into (4.1), because it contains no zeros on the real axis. The smeared distribution will however fall off rapidly away from the support of the large $N$ distribution. Therefore, the necessary zeros can be introduced by shifting the entire distribution slightly downwards, so that

$$n_N = \frac{1}{\pi i} \oint_{\Gamma_N} \frac{\rho_N'(z)}{\rho_N(z) - \eta_N} dz. \qquad (4.3)$$

The small number $\eta_N > 0$ and the contour $\Gamma_N$ will be specified shortly. The objective is to produce a well-defined quantity $n_N$ such that $\lim_{N \to \infty} n_N = n$. This will allow the topological integer $n$ to be extracted from numerics at large but finite $N$. In particular, it allows the critical temperature — where $n$ jumps — to be estimated systematically from finite $N$ numerics.

To choose the appropriate $\epsilon_N, \eta_N$ and $\Gamma_N$, we must understand the location of the poles and the zeros of the smeared distribution $\rho_N(z)$ in (4.2) as a function of $N$. It is easy to see that all zeros and poles of (4.2) are at least a distance $\epsilon_N$ away from the real axis. If, then, the contour $\Gamma_N$ runs above and below the real axis at a distance $\epsilon_N/2$, the only contribution to (4.3) is from zeros of $\rho_N(z) - \eta_N$ on the real axis. We must now define $\epsilon_N$ and $\eta_N$ so that these zeros only occur close to the boundaries of the large $N$ distribution $\rho(z)$.

As $N \to \infty$, the typical spacing between singular values, with our normalization, is $\Delta\lambda \sim N^{-1}$. So long as $\epsilon_N \gg \Delta\lambda$, so that the individual spikes associated with each singular value are smeared out, the distribution $\rho_N(z)$ should uniformly approach the large $N$ distribution $\rho(z)$. We will take $\epsilon_N = 2\,\mathrm{IQR}/\sqrt[3]{N}$, corresponding to the Freedman-Diaconis rule for binning. Here, IQR is the interquartile range.

The uniform convergence of the distribution breaks down at the boundaries of the distribution. Expanding (4.2) in $\epsilon_N$ and then taking the large $N$ limit, we can write

$$\rho_N(z) = \rho(z) - \frac{2\epsilon_N}{\pi} \omega'(z) + \cdots. \qquad (4.4)$$

Here, $\omega(z) = \sum_a (z - \lambda_a)^{-1}$ is the resolvent. We know that $\rho(z) \sim \sqrt{\lambda_\star - z} \sim 1/\omega'(z)$ close to a boundary $\lambda_\star$ of the distribution. Therefore, the correction in (4.4) is only small if $\epsilon_N \ll |\lambda_\star - z|$. Essentially this is because there are a large number of singular values accumulating close to the boundary of the distribution, and hence at such values of $z$ it is not legitimate to expand (4.2) in $\epsilon_N$. An accurate approximation to the endpoints can be found by taking a large enough shift $\eta_N$ so that $\epsilon_N \ll \eta_N^2$. However, this scaling overestimates the transition temperature at finite $N$ for the following reason. As the transition is approached from above, the distribution vanishes at an interior point $x_c$ as $\rho(z) \sim (z - x_c)^2$. The uniform convergence does not break down close to this smoother vanishing and taking a large $\eta_N$ introduces zeros at a higher temperature than necessary. To accurately capture the topological transition temperature, we can take instead a smaller shift, $\eta_N \sim \epsilon_N$. While this shift causes the location of outer

boundaries of the distribution to be incorrectly identified (for the reason just discussed), this fact does not matter for the topological quantity (4.3). Nothing interesting is happening with the outer boundaries.

Numerical simulations of the Ising model at some given finite $N$ produce many eigenvalues $\lambda_a$. By combining several independent Monte Carlo states, we can produce higher quality statistics for the thermally averaged distribution. Given the eigenvalues, we then choose $\eta_N \sim \epsilon_N$ as specified above and numerically find the zeros of the denominator of (4.3). The number of these zeros determines $n_N$. The results are shown in Fig. 3.

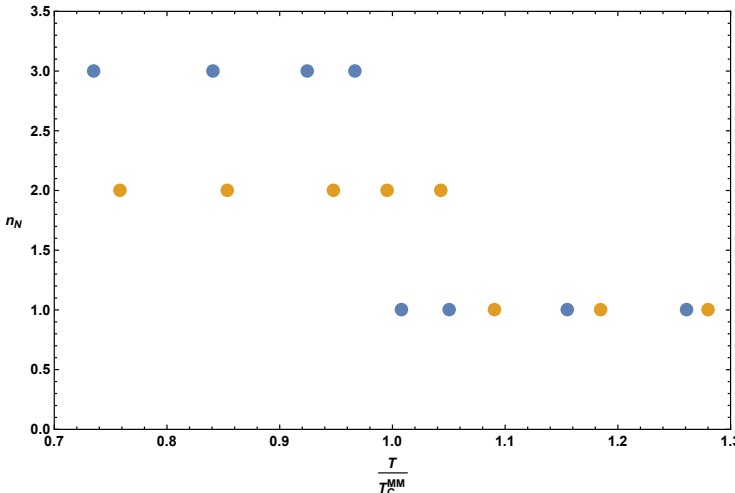

Figure 3: The finite N numerical topological index $n_N$ is plotted against a dimensionless temperature $T/T_C^{MM}$ for the same models as considered previously: blue dots are $(\nu_4 = -3, \nu_5 = 1)$ and yellow dots are $(\nu_3 = 1)$. Here, $T_c^{MM}$ is the transition temperature of the corresponding matrix integral. The total number of eigenvalues is $N_{\text{eff}} = 16800$, which is the rank of the matrix ($N = 120$) multiplied by twice the number of Monte Carlo sweeps (the factor of two comes from symmetrizing the distribution). We set $\epsilon_N = \eta_N = \epsilon_{FD}$ (the value prescribed by the Freedman Diaconis rule) for both parameter sets and obtain estimates of $T_c$ that are within 5% of the true $N \to \infty$ answer.

## 4.3   Topology competes with glassiness and magnetic order

The matrix description of the spin model does not hold at all temperatures. At low temperatures, the spins can either enter a glassy [1] or magnetically ordered [7] state, neither of which is captured by large $N$ matrices. If the transition to glassiness or ordering occurs at a temperature above the topological transition temperature $T_c$, then the topological transition is not realized in the spin model. We will now explain why these phases are outside of the matrix description and determine when they arise.

### 4.3.1   Magnetic order

When the potential $\hat{V}(x)$ is unbounded below, i.e. if the highest order term in the polynomial has a negative coefficient, the singular values want to fly off to infinity. While the constraint $\int dx \rho(x) x^2 = 1$ doesn't allow this, a solution to this constraint equation ceases to exist below a critical temperature. However, this fact is pre-empted by a first order phase transition at much higher temperatures, in which magnetically ordered configurations of Ising spins dominate the spin partition function [7]. Specifically, consider a matrix of Ising spins with all spins up:

$S_{aB} = +1$. From (1.2), this configuration has energy (let $-|v_n|$ be the coefficient of the highest order term)

$$E_0 = -|v_n| N^{n+1}. \tag{4.5}$$

The contribution of the lower order terms in the potential is subleading at large $N$. Due to the invariance of the Hamiltonian under flipping rows or columns of spins, there are $g_0 = 2^{2N-1}$ matrices of spins with the same energy. These configurations can be contrasted with the order $2^{N^2}$ matrix integral configurations which have energies of order $N^2$. The first order large $N$ transition therefore occurs at

$$T_{\mathrm{mag}} \sim |v_n| N^{n-1} \log 2, \tag{4.6}$$

where the low energy (4.5) overcomes the higher entropy of the matrix integral configurations. While these low energy states do not all have conventional ferromagnetic order, they can be characterized by a certain correlation between four spins at arbitrary separation [7].

It is clear that the magnetically ordered states are outside of the matrix integral saddle point. The ordered spin matrices have a single nonzero singular value of order $N$, in contrast to the singular values of order $\sqrt{N}$ in the matrix integral. This is the simplest way in which the spin softening in §2 can break down: there can be alternate configurations in the Ising partition function that dominate over the self-consistent matrix integral saddle. It is easy, however, to avoid this ordering by considering potentials that are bounded below.

### 4.3.2 Glassiness

A more ubiquitous breakdown of spin softening occurs due to glassiness at low temperatures [1]. The manifestation of glassiness in Fig. 1 was that the energy of the Ising configurations ceases to vary with temperature below some $T_{\mathrm{gl}}$. The distribution of singular values is also seen to freeze below this temperature. The interpretation of the glassy transition in the spin model is therefore the familiar one: the energy landscape is extremely complex at low temperatures, with many local minima, and the system becomes trapped in a metastable minimum. In the matrix description, in contrast, we see as in Fig.1 that the energy curve continues smoothly down to a lower energy. This difference in behaviors is possible because the matrices are valued in a hyperspherical configuration space, $M_{aB} \in S^{N_1 N_2 - 1}$, while the spin configurations take values among the discrete $2^{N_1 N_2}$ vertices of an $N_1 N_2$-dimensional hypercube. Glassy configurations which are local minima in the discrete space of spins need not be local minima on the sphere: there can be 'easy' directions or 'valleys' along which the free energy can be decreased towards the global minimum.[2]

For models with a monomial potential $\hat{V}(x) = v_n x^{2n}$, the onset of glassiness at $T_{\mathrm{gl}}$ occurs below the topological transition temperature $T_c$ in (3.11) only for $n = 2$ and $n = 3$. For $n = 2$ we find $T_{\mathrm{gl}} \approx 0.5$, while $T_c = 1$. For $n = 3$, $T_{\mathrm{gl}} \approx 1.2 - 1.5$ while $T_c \approx 2.1$. For $n \geq 4$, we find $T_{\mathrm{gl}} > T_c$ and hence the singular value distribution freezes before disconnecting, and there is no topological transition. The two temperatures are quite close for $n = 4$, but become increasingly different at larger $n$. For example, for $n = 6$ we find $T_{\mathrm{gl}} \approx 10 - 15$ while $T_c \approx 8.7$. For $n = 8$, $T_{\mathrm{gl}} \approx 22 - 30$, while $T_c \approx 18.9$. The range of quoted values for $T_{\mathrm{gl}}$ comes from finite $N$ uncertainties in simulations with $N = 100$.

For polynomial potentials with local minima away from the origin, such as (3.20), a topological transition can be induced at arbitrarily high temperatures by having a strongly negative term in the potential. These negative terms do not favor glassiness (on the contrary, they lead to a sort of local magnetic ordering, as we see in the following subsection), and therefore lead to a large class of models where a large $N$ topological transition occurs. We saw an example of such a transition in Fig. 2.

---

[2]We thank Daniel Ranard for this intuitive picture.

## 4.4 Exact ground states

Following [1, 13] we can establish in certain cases that the minimum energy attained by the matrix integral is in fact the ground state energy of the spin system. In these cases the exact ground states can be constructed, despite the presence of glassiness. When $N = 2^k$, with $k$ integral, one can easily construct matrices of spins $S_\perp$ with mutually orthogonal rows:

$$\left(S_\perp S_\perp^T\right)_{ab} = N\delta_{ab}. \tag{4.7}$$

The energy of such a configuration with the Hamiltonian (1.2) is immediately evaluated as $E = N^2\hat{V}(1)$. This agrees with the zero temperature matrix energy found in §3.2 for potentials with a single minimum at the origin. Indeed, it is the ground state energy of the spin system when all of the coefficients in the potential $v_n \geq 0$. This is because the energy of the configuration attains the lower bound $\frac{1}{N^{n-1}}\mathrm{tr}\left[(SS^T)^n\right] \geq \frac{1}{N^{n-1}}\sum_a[(SS^T)_{aa}]^n = N^2$. In the last step we used the fact that each spin squares to 1. We will see what happens when one or more of the $v_n$ are negative shortly. Finally, the spin matrices $S_\perp$ obeying 4.7 have singular values $\pm\sqrt{N}$ and therefore correspond to the low temperature distribution $\rho(x) = \frac{1}{2}(\delta(x-1) + \delta(x+1))$ described below (3.7) above.

The above construction can be generalized to the case when some terms in the potential are negative. Consider for example potentials of the form (3.20). We would like to construct matrices of spins with the singular value distribution (3.21). Let $S_\parallel$ be a $2^l \times 2^l$ dimensional matrix with entries all equal to one, similar to the magnetically ordered matrices we considered in §4.3.1. Now let $S_\perp$ be a $2^{k-l} \times 2^{k-l}$ dimensional matrix with mutually orthogonal rows as in (4.7). Here $k \geq l$. Then construct the $N \times N$ matrix, with $N = 2^k$,

$$S = S_\parallel \otimes S_\perp. \tag{4.8}$$

The matrix $SS^T$ is then seen to be block diagonal, with $2^{k-l}$ blocks each given by the $2^l \times 2^l$ dimensional matrix $S_\parallel S_\parallel^T$, times the number $2^{k-1}$. This matrix has eigenvalue 0 with multiplicity $N - 2^{k-l}$ and singular value $2^{k+l}$ with multiplicity $2^{k-l}$. Thus we obtain the distribution (3.21) with $2s_\star = 2^{-l}$ and $x_\star = 2^{l/2}$. As in §3.3, $s_\star$ and hence $l$ are to be determined by minimizing the energy on this set of configurations.

These microscopic configurations give a sense of what the matrix Ising model 'wants' to do at low temperatures. Loosely put, positive terms in the potential are minimized by spin matrices with orthogonal rows, while negative terms are minimized by highly degenerate matrices. A balance between these two tendencies is achieved with tensor product matrices such as (4.8). For general $N \neq 2^k$, orthogonality cannot be perfectly realized. In all cases where magnetic ordering does not occur, a glassy phase intervenes and these exact low energy states cannot be reached. For example, for $N = 2^4$, the Markov chain Monte Carlo algorithm we are employing finds the true ground state for monomial potentials with $n = 2, 3, 4$. For $N = 2^5$ however, the algorithm fails to find the true ground state. This supports the intuition that the glassy states become long-lived in the large $N$ limit. In fact, by increasing $n$ at fixed $N = 2^5$, we observe a parametric growth in the energy of the glassy states, even though the true ground state remains at $E = v_n N^2$.

## 5 Discussion

Spin softening describes the emergence of continuous degrees of freedom from an underlying discrete dynamics. It is believed that an analogous phenomenon underpins several important aspects of gravitational physics, most notably the finiteness of the Bekenstein-Hawking black hole entropy. Systems in which the self-erasure of discreteness can be demonstrated explicitly

can serve as useful toy models for this physics. Indeed, the steps in the spin softening theorem of §2 — in particular the introduction of collective fields built from the spins — have some similarity to those used in solving the SYK model for black hole dynamics (e.g. [14]). This is not a coincidence, as both have a common ancestry in methods used to study spin glasses [8,15].

To connect more deeply with gravitational dynamics, it would be necessary to extend these methods to quantum spin systems. Some first steps in this direction were taken in [9]. It was found that a straightforward generalization of the spin softening does not go through in the quantum case. We explain this in Appendix C, where we show how several of the steps in the spin softening logic can be adapted to the quantum case. In a nutshell, the problem is the following: The propagator $P$ of the $w$ fields in §2 is bilocal in time in the quantum case, $P_{ab}(t, t')$, and the corresponding constraint removing the non-singlet terms is also bilocal in time. However, the variables $\mu_a$ only depend on a single time. There is not enough freedom in the $\mu_a(t)$ functions to satisfy the bilocal in time constraints.

Nonetheless, softening in quantum spin systems is ubiquitous at continuous quantum critical points [16]. Such critical points are characterized by the presence of many excitations at energies parametrically below the microscopic spin flip energy scale. The 'slow' dynamics of these degrees of freedom is often described by continuous quantum mechanical theories. This brings us to the main topic of our paper, which is the existence of topological phase transitions in matrix Ising models. It is well known that phase transitions in matrix quantum mechanics are associated to emergent gapless degrees of freedom. That fact underpins the emergence of spacetime in lower dimensional string theories [17].

In [9] a matrix quantum mechanics theory was proposed to describe the critical excitations near a quantum topological transition in a transverse field matrix Ising system. However, it was not shown that this topological transition actually occurred in the model studied. The main complication is the presence of competing glassy phases, as in the classical models we have discussed in this paper. However, in this paper we have understood how, by extending the spin softening theorem to a larger family of matrix Ising models, the topological transition can be favored over glassiness. It is of interest to revisit the quantum systems, perhaps together with quantum Monte Carlo simulations, to identify a quantum critical point within this class of theories.

Finally, in a different direction, there are rich connections between matrix dynamics, string theory and the geometry of Riemann surfaces (e.g. [18,19]). The integer (4.1) is an impoverished proxy for the genus of a Riemann surface associated to the distribution of singular values. It is possible that a more thorough connection to those ideas will reveal a richer topological structure in the different phases of the large $N$ matrix Ising models, as is common in other instances of topological order [20].

## Acknowledgements

We acknowledge helpful discussions with Jordan Cotler, Ilya Esterlis, Xizhi Han, Jonathan Luk, Daniel Ranard, Phil Saad, Michail Savvas, Stephen Shenker, Umut Varolgunes. We also thank Yibing Du for helpful comments on the draft. The work of SAH is partially supported by DOE award DE-SC0018134. ZDS is supported by the Physics/Applied Physics/SLAC Summer Research Program for undergraduates at Stanford University.

## A Proof of $\sigma$-Propagator Scaling

It was crucial to the proof of spin softening in §2 to understand the $N$ scaling of $\sigma$. Since our action is not quadratic in $G$, we cannot analytically integrate it out to obtain the effective action for $\sigma$ and hence read off its $N$ scaling. In this Appendix we will obtain the effective action order by order in $\sigma$. For simplicity we will work with the case of a monomial potential $V = \frac{v_n}{N^{n-1}} \mathrm{tr}\big(SS^T\big)^n$.

To make analytic progress on the $G$ integral, we add and subtract a Gaussian term $-\frac{1}{2} \frac{(\beta v_n)^c}{N^d} \mathrm{tr}\, G^2$, and make a shift to $\tilde{G} = G - \frac{iN^d}{(\beta v_n)^c}\sigma$. We pick $c = \frac{2}{n}$ and $d = 1$ so that the moments of $G$ under the the weighting $e^{-\frac{\beta v_n}{N^{n-1}} \mathrm{tr}\, G^n}$ have the same scaling with $\beta v_n, N$. With this choice of $c, d$, we can rewrite the partition function as

$$
\begin{aligned}
Z &= \int dG\, e^{-\frac{\beta v_n}{N^{n-1}} \mathrm{tr}\, G^n + i\, \mathrm{tr}\, G\sigma} = \int dG\, e^{-\frac{1}{2}\frac{(\beta v_n)^{2/n}}{N} \mathrm{tr}\, G^2 + i\, \mathrm{tr}\, G\sigma} e^{\frac{1}{2}\frac{(\beta v_n)^{2/n}}{N} \mathrm{tr}\, G^2 - \frac{\beta v_n}{N^{n-1}} \mathrm{tr}\, G^n} \\
&= e^{-\frac{1}{2}\frac{N}{(\beta v_n)^{2/n}} \mathrm{tr}\, \sigma^2} \int dG\, e^{-\frac{1}{2}\frac{(\beta v_n)^{2/n}}{N} \mathrm{tr}\left(G - \frac{iN}{(\beta v_n)^{2/n}}\sigma\right)^2} e^{\frac{1}{2}\frac{(\beta v_n)^{2/n}}{N} \mathrm{tr}\, G^2 - \frac{\beta v_n}{N^{n-1}} \mathrm{tr}\, G^n} \\
&= e^{-\frac{1}{2}\frac{N}{(\beta v_n)^{2/n}} \mathrm{tr}\, \sigma^2} \int d\tilde{G}\, e^{-\frac{1}{2}\frac{(\beta v_n)^{2/n}}{N} \mathrm{tr}\, \tilde{G}^2} e^{\frac{1}{2}\frac{(\beta v_n)^{2/n}}{N} \mathrm{tr}\left(\tilde{G} + \frac{iN}{(\beta v_n)^{2/n}}\sigma\right)^2 - \frac{\beta v_n}{N^{n-1}} \mathrm{tr}\left(\tilde{G} + \frac{iN}{(\beta v_n)^{2/n}}\sigma\right)^n}.
\end{aligned}
\tag{A.1}
$$

For simplicity of notation, define new variables

$$
A = \frac{1}{2}\frac{(\beta v_n)^{2/n}}{N} \mathrm{tr}\left(\tilde{G} + \frac{iN}{(\beta v_n)^{2/n}}\sigma\right)^2, \quad B = -\frac{\beta v_n}{N^{n-1}} \mathrm{tr}\left(\tilde{G} + \frac{iN}{(\beta v_n)^{2/n}}\sigma\right)^n.
\tag{A.2}
$$

In terms of $A, B$, the $\tilde{G}$ integral takes a nice form that facilitates standard Feynman diagram calculations:

$$
\begin{aligned}
\log Z &= -\frac{1}{2}\frac{N}{(\beta v_n)^{2/n}} \mathrm{tr}\, \sigma^2 + \sum_{m=1}^{\infty} \frac{1}{m!}\left\langle (A+B)^m \right\rangle_c \\
&= -\frac{1}{2}\frac{N}{(\beta v_n)^{2/n}} \mathrm{tr}\, \sigma^2 + \sum_{m=1}^{\infty}\sum_{k=0}^{m} \frac{1}{m!}\binom{m}{k}\left\langle A^k B^{m-k}\right\rangle_c.
\end{aligned}
\tag{A.3}
$$

The connected diagrams are in general tedious to compute. But fortunately we only care about the scaling of these diagrams with $N, \beta v_n$. We warm up by computing these scalings in the $m = 1$ term:

$$
\langle A \rangle_c = \left\langle \frac{1}{2}\frac{(\beta v_n)^{2/n}}{N} \mathrm{tr}\, \tilde{G}^2 \right\rangle_c + \left\langle \frac{1}{2}\frac{(\beta v_n)^{2/n}}{N} \cdot \frac{i^2 N^{2d}}{(\beta v_n)^{2c}} \mathrm{tr}\, \sigma^2 \right\rangle_c = \frac{1}{2}N^2 - \frac{N}{2(\beta v_n)^{2/n}} \mathrm{tr}\, \sigma^2,
\tag{A.4}
$$

$$\langle B\rangle_c = \left\langle \frac{\beta v_n}{N^{n-1}}\operatorname{tr}\tilde{G}^n\right\rangle_c + \binom{n}{2}\left\langle \frac{\beta v_n}{N^{n-1}}\operatorname{tr}\tilde{G}^{n-2}(\frac{iN}{(\beta v_n)^{2/n}}\sigma)^2\right\rangle_c$$
$$+ \binom{n}{4}\left\langle \frac{\beta v_n}{N^{n-1}}\operatorname{tr}\tilde{G}^{n-4}(\frac{iN}{(\beta v_n)^{2/n}}\sigma)^4\right\rangle + O(N,\sigma^6)$$
$$= \frac{\beta v_n}{N^{n-1}}\cdot N^{n/2}\cdot c_n\cdot N^{n/2+1} + \frac{\beta v_n}{N^{n-1}}(\frac{N}{(\beta v_n)^{2/n}})^{(n-2)/2}N^{(n-2)/2}\cdot c_{n-2}\binom{n}{2}\cdot(-\frac{N^2}{(\beta v_n)^{4/n}}\operatorname{tr}\sigma^2)$$
$$+ \frac{\beta v_n}{N^{n-1}}(\frac{N}{(\beta v_n)^{2/n}})^{(n-4)/2}N^{(n-4)/2}\cdot c_{n-4}\binom{n}{4}\frac{N^4}{(\beta v_n)^{8/n}}\operatorname{tr}\sigma^4 + O(N,\sigma^6)$$
$$= c_n\beta v_n N^2 - c_{n-2}\binom{n}{2}(\beta v_n)^{-2/n}N\operatorname{tr}\sigma^2$$
$$+ c_{n-4}\binom{n}{4}(\beta v_n)^{-4/n}N\operatorname{tr}\sigma^4 + O(N,\sigma^6).$$

(A.5)

In the calculation above, $c_n = \frac{1}{n+1}\binom{2n}{n}$ denotes the Catalan number counting the number of planar diagrams at a given order.

Two observations can be made at this point. First of all, we can prove that all terms proportional to $\operatorname{tr}\sigma^2$ in the effective action come with a prefactor $N$. This is because in the expansion of $\log Z$, a term like $\left\langle A^k B^{m-k}\right\rangle_c$ contributes to $\operatorname{tr}\sigma^2$ in three ways:

(1) One factor of $\sigma$ in $A^k$ and one factor of $\sigma$ in $B^{m-k}$.
(2) Two factors of $\sigma$ in $A^k$ and no factor of $\sigma$ in $B^{m-k}$.
(3) No factor of $\sigma$ in $A^k$ and two factors of $\sigma$ in $B^{m-k}$.

In the previous computation, we have already shown that at lowest order, cases (2) and (3) give the correct $N$ scaling. At higher orders, one can check explicitly that the scaling doesn't change. The calculation is not very enlightening, so we will not include it. Case (1), however, appears for the first time in $m=2$, where we have the cross term $\langle AB\rangle_c$. Now let's investigate how $\langle AB\rangle_c$ generates something proportional to $\operatorname{tr}\sigma^2$:

$$\langle AB\rangle_c = \left\langle (\frac{1}{2}\frac{(\beta v_n)^{2/n}}{N}\operatorname{tr}\tilde{G}^2)\cdot(-\frac{\beta v_n}{N^{n-1}}\operatorname{tr}\tilde{G}^{n-2}(\frac{iN}{(\beta v_n)^{2/n}}\sigma)^2)\right\rangle_c$$
$$\propto (\beta v_n)^{1-2/n}N^{2-n}\cdot(\frac{N}{(\beta v_n)^{2/n}})^{n/2}\cdot N^{n/2-1}\operatorname{tr}\sigma^2$$

(A.6)

$$= (\beta v_n)^{-2/n}N\operatorname{tr}\sigma^2 = \frac{N}{(\beta v_n)^{2/n}}\operatorname{tr}\sigma^2.$$

Notice that because we have $n$ factors of $\tilde{G}$, we get $n/2$ propagators $(\frac{N}{\beta v_n})^{n/2}$. The factor of $N^{n/2-1}$ comes from the $n/2-1$ loops[3]. We therefore recover the same $N,\beta v_n$ scaling as in cases (2) and (3).

The second observation concerns higher order terms in the effective action. Given that the Gaussian part of the action goes as $-N\operatorname{tr}\sigma^2$, we claim that all terms involving $\operatorname{tr}\sigma^k$ must come with a prefactor of $N$ in order for the free energy to be extensive. For example, using $-N\operatorname{tr}\sigma^2$ as the quadratic term, $\left\langle N\operatorname{tr}\sigma^4\right\rangle \propto N\cdot(\frac{1}{N})^2\cdot N^3 = N^2$ where $(\frac{1}{N})^2$ comes from two powers of the propagator, and $N^3$ comes from the three loops in the diagram. This combinatorial pattern remains true for all $k$, thus validating our claim.

Using these observations, we can establish the form of the effective action and the desired scaling of the $\sigma$ propagator:

---

[3]Note that this is different from the usual $n/2+1$ loops. The difference of 2 comes from the fact that the two summation indices in $\operatorname{tr}\sigma^2$ cannot be pulled out of the trace.

**Proposition A.1.** *The effective action for $\sigma$ generated from the $\tilde{G}$ integral has the following structure:*

$$\log Z = C - \frac{N}{(\beta v_n)^{2/n}} \operatorname{tr} \sigma^2 \cdot F_2(n) + F_4(n)\frac{N}{(\beta v_n)^{4/n}} \operatorname{tr} \sigma^4 + \ldots + F_{2k}(n)\frac{N}{(\beta v_n)^{2k/n}} \operatorname{tr} \sigma^{2k}. \quad \text{(A.7)}$$

*Where $C$ comes from resumming all the $\sigma$-independent terms in the perturbative expansion. In addition, the dressed propagator under the full effective action has the same $\beta v_n, N$ scaling as the bare propagator.*

*Proof.* The form of the effective action follows directly from the scaling of connected diagrams $\left\langle A^k B^{m-k} \right\rangle_c$ that we have already established. The only new thing that we need to check is that the dressed propagator for $\sigma$ generated by the effective action always scales as $\frac{(\beta v_n)^{2/n}}{N}$.

Suppose we expand the non-Gaussian terms in the effective action. Then when we calculate the propagator $\left\langle \sigma_{ij} \sigma_{kl} \right\rangle$, we encounter terms like:

$$\left\langle F_{2k}(n)\frac{N}{(\beta v_n)^{2k/n}} \operatorname{tr} \sigma^{2k} \sigma_{ij} \sigma_{kl} \right\rangle. \quad \text{(A.8)}$$

Since the bare propagator is $\frac{(\beta v_n)^{2/n}}{N}$ and we have $k+1$ factors of the propagator, this term evaluates to something proportional to $\left(\frac{(\beta v_n)^{2/n}}{N}\right)^{k+1} \cdot \frac{N}{(\beta v_n)^{2k/n}} \cdot N^{k-1}$ where $k-1$ is the number of loops. Therefore:

$$\left\langle F_{2k}(n)\frac{N}{(\beta v_n)^{2k/n}} \operatorname{tr} \sigma^{2k} \sigma_{ij} \sigma_{kl} \right\rangle \propto \delta_{ik}\delta_{jl} \frac{(\beta v_n)^{2/n}}{N}. \quad \text{(A.9)}$$

The same proof works for any term that can appear in the expansion (cross terms can be handled in a similar way). Thus, the dressed propagator has the same N scaling as the bare propagator. $\qquad \square$

In conclusion, the effective action for $\sigma$ generates a dressed propagator that scales as $\frac{(\beta v_n)^{2/n}}{N}$. This is precisely the scaling needed to satisfy the first part of condition (2.5).

# B  A distinct class of Hamiltonians

In this Appendix we show that the steps in §2 can be adapted to a distinct class of Hamiltonians:

$$H = \sum_n \frac{u_n}{N^n} \sum_{a \neq b} [(SS^T)_{ab}]^{2n} = U(SS^T). \quad \text{(B.1)}$$

After introduction of $G, \sigma$ fields, we can rewrite the partition function in a form similar to that appearing in (2.1):

$$Z(\beta) = \int dG\, e^{-\beta U(G)} \int \frac{d\sigma}{2\pi} e^{-i \operatorname{tr} \sigma G} \operatorname{Tr} e^{i \operatorname{tr} \sigma SS^T}. \quad \text{(B.2)}$$

The final term here is the same as for the model considered in the main text, and hence (2.2) can again be used. It remains to establish the scaling of $\sigma$ that follows from the $G$ integral. We will now see that this scaling is the same as for the previous model.

The $G$ integral factorizes because

$$\int dG\, e^{-\beta U(G)} e^{-i \operatorname{tr} \sigma G} = \prod_{a \neq b} \left[ \int dG_{ab}\, e^{-\beta \sum_n \frac{u_n}{N^n}(G_{ab})^{2n}} e^{-i \sigma_{ab} G_{ab}} \right]. \quad \text{(B.3)}$$

To extract the $N$ scaling of various quantities, we can define new variables $\bar{G}_{ab} = \frac{G_{ab}}{\sqrt{N}}$ and $\bar{\sigma}_{ab} = \sqrt{N}\sigma_{ab}$, so that the integral for each factor simplifies to

$$\int dG_{ab} e^{-\beta \sum_n \frac{u_n}{N^n}(G_{ab})^{2n}} e^{-i\sigma_{ab}G_{ab}} = \sqrt{N} \int d\bar{G}_{ab} e^{-\sum_n \beta u_n(\bar{G}_{ab})^{2n} - i\bar{\sigma}_{ab}\bar{G}_{ab}}. \tag{B.4}$$

Treating the last line as an effective action for $\sigma_{ab}$, we can compute the mean and variance of $\sigma$ under the effective action. By symmetry the mean is zero, while the variance

$$(\Delta\sigma_{ab})^2 = \left\langle \sigma_{ab}^2 \right\rangle = \int d\bar{G}_{ab} e^{-\sum_n \beta u_n(\bar{G}_{ab})^{2n}} \frac{1}{N} \int d\bar{\sigma}_{ab} \bar{\sigma}_{ab}^2 e^{-i\bar{\sigma}_{ab}\bar{G}_{ab}} \sim \frac{1}{N}. \tag{B.5}$$

This is the same scaling for $\Delta\sigma_{ab}$ as for the model considered in the main text, and hence the condition (2.5) that needs to be imposed to drop the non-singlet terms (in the $w$ integral) is the same.

The remaining steps all proceed as in §2, again leading to

$$Z(\beta) = \left(2e^{-\frac{1}{2}}\right)^{N_1 N_2} \int dM \int d\mu e^{i\mu[N_1 N_2 - \text{tr}(MM^T)]} e^{-\beta \text{tr}[U(MM^T)]}, \tag{B.6}$$

where now $U$ is given by (B.1). We have verified (B.6) numerically for this class of models, by matching the energies as a function of temperature (analogously to Fig. 1). It is worth noting that this family of models does not have an emergent $O(N_1, \mathbb{R}) \times O(N_2, \mathbb{R})$ symmetry. This means that the matrix integral cannot be solved using standard techniques. Nonetheless it was important that the Ising model still had an $O(N_1, \mathbb{Z}) \times O(N_2, \mathbb{Z})$ symmetry.

## C  Remarks on Quantum Generalizations

The transverse field matrix Ising Hamiltonian is

$$H = H_0 + \text{tr}\left[V(S^z S^{zT})\right], \qquad H_0 = -h \sum_{aB} S_{aB}^x. \tag{C.1}$$

Here $V$ is as in (1.2) in the main text. The quantum disordering transverse field term $H_0$ has been added at each site. This term preserves the symmetries of the classical Ising model [9].

To obtain a path integral expression for the partition function, a Suzuki-Trotter decomposition can be used. The Euclidean time direction is divided into $M$ segments of length $\epsilon = \beta/M \ll 1$. In terms of a basis of states $|S\rangle$ that are eigenvectors of $S_{aB}^z$ one has:

$$
\begin{aligned}
Z = \text{Tr}\, e^{-\beta H} &= \sum_{S_{aB}(m)} \prod_{m=1}^{M} \langle S(m)| e^{-\epsilon H_0} e^{-\epsilon V} |S(m+1)\rangle \\
&= e^{-MJ} \sum_{S_{aB}(m)} \exp\left\{ \sum_{m=1}^{M} \sum_{a,B} J S_{aB}(m) S_{aB}(m+1) \right\} \cdot \exp\left\{ -\epsilon \sum_{m=1}^{M} V[(SS^T)(m)] \right\}.
\end{aligned}
\tag{C.2}
$$

Where $J = -\frac{\log(\epsilon h)}{2}$ is proportional to the effective energy cost of a spin flip (this term is obtained in a standard way by expanding $e^{-\epsilon H_0}$ to first order in $\epsilon$) and $\sum_{S_{aB}(m)}$ denotes the sum over all spin configurations $S_{aB}(m) = \pm 1$.

Introducing $G, \sigma$ fields for the spin variables $S_{aB}(m)$ yields, performing manipulations similar to in §2,

$$Z \propto \int DGD\sigma \exp\left\{-\epsilon \sum_{m=1}^{M} \operatorname{tr} V[G(m)] - i\epsilon \sum_{m=1}^{M} \operatorname{tr} \sigma(m)G(m) + i\epsilon \sum_{m=1}^{M} \mu(m)N^2\right\}$$
$$\cdot \left(\sum_{S_a(m)} \exp\left\{i\epsilon \sum_{m=1}^{M} \sum_{ab}\left[(\sigma_{ab}(m) - \mu(m)\delta_{ab})S_a(m)S_b(m) + \frac{J}{\epsilon}\delta_{ab}S_a(m)S_b(m+1)\right]\right\}\right)^{N_2},$$
(C.3)

where in the last line we again factorized the spin trace utilizing the $O(N_2, \mathbb{Z})$ symmetry of the Hamiltonian, so that there is only a sum over spins $S_a \equiv S_{a1}$. We have also directly set all the $\mu(m)_a = \mu(m)$ equal. Note that these undetermined quantities now depend on $m$.

Define the term inside the final bracket in (C.3) as $z(\sigma, \mu)$. After introducing a new variable $\tilde{\sigma}_{ab} = 2i(\sigma_{ab} - \mu\delta_{ab})$ and doing a Hubbard Stratonovich transformation on $z(\sigma, \mu)$, we can further factorize the trace over Ising variables as in equation (2.3) and (2.4) of the main text:

$$z(\sigma, \mu) = \frac{1}{\prod_m \sqrt{\det \tilde{\sigma}(m)}} \int Dw \exp\left\{\frac{\epsilon}{2}\sum_{m,a,b}[w_a(\tilde{\sigma}^{-1})_{ab}w_b](m)\right\}\prod_a z_a(w, J), \qquad (C.4)$$

where for each $a$, $z_a(w, J)$ is the partition function of a 1D classical Ising model with $M$ sites and periodic boundary conditions $S_a(1) = S_a(M+1)$:

$$z_a(w, J) = \sum_{S_a(m)} \exp\left\{-\epsilon \sum_{m=1}^{M}\left(w_a(m)S_a(m) + \frac{J}{\epsilon}S_a(m)S_a(m+1)\right)\right\}. \qquad (C.5)$$

At this point, we would like to obtain an expression analogous to (2.4) in the main text, and then argue that the higher order in $w$, non-singlet, terms can be dropped by some suitable choice of $\mu$. To this end we expand the first term $e^{-\epsilon \sum_{m=1}^{M} w_a S_{a1}(m)}$ in (C.5), evaluate the spin traces using the exact correlation functions of the 1D Ising model with interaction $JS_a(m)S_a(m+1)$, and then re-exponentiate. This leads to

$$z_a(w, J) = \exp\left\{\frac{\epsilon^2}{2}\sum_{m,m'} w_a(m)K(m - m')w_a(m') + \text{non-singlets}\right\}. \qquad (C.6)$$

Here the propagator

$$K(m - m') = e^{|m' - m|\log\tanh J}. \qquad (C.7)$$

Using (C.6) in (C.4), and taking the continuum limit $\epsilon \to 0$ with time $t = \epsilon m$ fixed, the effective path integral takes a form that is reminiscent of (2.4):

$$z(\sigma, \mu) \propto \frac{1}{\sqrt{\det \tilde{\sigma}}} \int Dw \exp\left\{-\frac{1}{2}\int dt dt' w_a(t)(\tilde{\sigma}^{-1} - K)_{ab}(t, t')w_a(t') + \text{nonsinglets}\right\}. \qquad (C.8)$$

In the continuum limit,

$$K(t - t') = e^{-2\beta h|t - t'|}, \qquad (C.9)$$

is the thermal propagator of a harmonic oscillator with thermal mass proportional to $h$. More precisely:

$$K^{-1} = -\frac{1}{8h\tanh(\beta h)}\frac{d^2}{dt^2} + \frac{h}{2\tanh\beta h}. \qquad (C.10)$$

Following manipulations in the main text, we can expand the nonsinglet terms in (C.8) and Wick contract using the propagator:

$$P_{ab}(t, t') = \left( \frac{1}{\tilde{\sigma}^{-1} - K} \right)_{ab} (t, t') = \left( \frac{\tilde{\sigma}}{1 - K\tilde{\sigma}} \right)_{ab} (t, t'). \tag{C.11}$$

A scaling argument similar to that used to establish (2.5) shows that for $a \neq b$

$$P_{ab}(t, t') \sim \frac{1}{\sqrt{N}}. \tag{C.12}$$

At this point, one would like to choose $\mu_a(t)$ so that for all $a, t, t'$:

$$P_{aa}(t, t') = 0. \tag{C.13}$$

If we can satisfy these constraints and thus drop the nonsinglet terms, we can obtain a matrix quantum mechanics by integrating back out the $G, \sigma$ fields, just as we did in §2. We would obtain

$$Z \propto \int DM \delta(N^2 - \text{tr}\, MM^T) \exp\left\{ -\int dt \left( \frac{1}{2} \text{tr}\left[ MK^{-1}M^T \right](t) + V[MM^T(t)] \right) \right\}. \tag{C.14}$$

Recall from (C.10) that $K^{-1}$ is a local in time operator, and so this is the partition function of a constrained matrix quantum mechanics.

It thus suffices to establish the constraint in (2.5). A priori, this seems impossible because $\mu(t)$ is a local in time Lagrange multiplier, while $P_{aa}(t, t') = 0$ is a bilocal constraint. However, for the case of $n = 2$ considered in [9] the expectation values $\langle P_{aa}(t, t') \rangle$ under the $G, \sigma$ path integral are time independent up to third order in $\beta v_4$, and the constraints $\langle P_{aa}(t, t') \rangle = 0$ are exactly enforced by the saddle point equations for $\mu(t)$ under the $\mu(t)$ path integral (analogously to what happened in the classical case in the main text). This low order 'miracle' explains the matching of ground state energies for the spin and matrix models to third order in perturbation theory, noted in [9]. Beyond third order, the constraint equations $\langle P_{aa}(t, t') \rangle = 0$ become genuinely bilocal in time and $\mu(t)$ no longer has enough degrees of freedom to enforce all the constraints.

This argument leaves open the hope that we can go in the reverse direction: start with a matrix quantum mechanics with some bilocal constraint, and adjust the constraint carefully to match the transverse field Ising model to all orders in perturbation theory. It turns out that this reverse direction is also impossible because the diagrammatic expansion for the transverse field Ising model involves integrals over multi-local functions in time. For example, at $2n$-th order in perturbation theory, one encounters a diagram involving the correlator $F(t_1, t_2, \ldots, t_{2n}) = \cosh(\beta h - 2\beta h|t_1 - t_2 + \ldots t_{2n-1} - t_{2n}|)$. This class of correlators fail to Wick factorize. On the other hand, diagrams for a matrix quantum mechanics with bilocal constraint always Wick factorize into products of propagators. This discrepancy between the two diagrammatic expansions makes it very hard for a single bilocal in time constraint equation to give free energy agreement for all values of the coupling constants (i.e. $\beta, v_n, h$).

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
