# Peer review of "Topological order in matrix Ising models"

_SciPost Physics, doi:SciPost Phys. 7, 081 (2019)_

## Round 1 · Referee Report · Anonymous (Referee 1) · 2019-11-22

Strengths

  1. clear and interesting message
  2. convincing analytic (matrix integrals) and numerical (Ising spins) analysis 3 discussion with wider context and outlook

Report

Central theme of this study is what the authors call “self-erasure of discreteness”. In the concrete model studied, this amounts to a collection of $N_1 N_2$ Ising spins being described by continuous bosonic degrees of freedom with a single constraint. With this spin softening in place, topological large $N$ phase transitions are expected, and the authors indeed establish them in various guises, depending on a choice of interaction potential. Importantly, the topological phase transitions happen at temperatures well above a glassy free-out transition, where the matrix integral loses its relevance for the problem.

The manuscript brings a very clear and interesting message and the underlying analysis (analytics for the matrix integrals and Monte Carlo simulations on the spin systems) is convincing.

The discussion in section 5 reveals some of the true motivations of the authors: a deeper connection with gravitational physics via an extension to the quantum case, building on ref [9] by (in part) the same authors.

---

## Editorial Decision

published